# Towards the Continuous Manufacturing of Liquisolid Tablets Containing Simethicone and Loperamide Hydrochloride with the Use of a Twin-Screw Granulator

**DOI:** 10.3390/pharmaceutics15041265

**Published:** 2023-04-18

**Authors:** Daniel Zakowiecki, Margarethe Richter, Ceren Yuece, Annika Voelp, Maximilian Ries, Markos Papaioannou, Peter Edinger, Tobias Hess, Krystyna Mojsiewicz-Pieńkowska, Krzysztof Cal

**Affiliations:** 1Chemische Fabrik Budenheim KG, Rheinstrasse 27, 55257 Budenheim, Germanytobias.hess@budenheim.com (T.H.); 2Thermo Electron (Karlsruhe) GmbH, Pfannkuchstrasse 10-12, 76185 Karlsruhe, Germany; margarethe.richter@thermofisher.com (M.R.); ceren.yuece@thermofisher.com (C.Y.);; 3Thermo Fisher Scientific GmbH, Im Steingrund 4-6, 63303 Dreieich, Germany; maximilian.ries@thermofisher.com; 4Department of Physical Chemistry, Faculty of Pharmacy, Medical University of Gdansk, al. Gen. J. Hallera 107, 80-416 Gdansk, Poland; krystyna.pienkowska@gumed.edu.pl; 5Department of Pharmaceutical Technology, Faculty of Pharmacy, Medical University of Gdansk, al. Gen. J. Hallera 107, 80-416 Gdansk, Poland; kcal@wp.pl

**Keywords:** continuous manufacturing, liquisolid tablets, simethicone, loperamide hydrochloride, twin-screw granulator, tribasic calcium phosphate

## Abstract

Continuous manufacturing is becoming the new technological standard in the pharmaceutical industry. In this work, a twin-screw processor was employed for the continuous production of liquisolid tablets containing either simethicone or a combination of simethicone with loperamide hydrochloride. Both active ingredients present major technological challenges, as simethicone is a liquid, oily substance, and loperamide hydrochloride was used in a very small amount (0.27% *w*/*w*). Despite these difficulties, the use of porous tribasic calcium phosphate as a carrier and the adjustment of the settings of the twin-screw processor enabled the optimization of the characteristics of the liquid-loaded powders and made it possible to efficiently produce liquisolid tablets with advantages in physical and functional properties. The application of chemical imaging by means of Raman spectroscopy allowed for the visualization of differences in the distribution of individual components of the formulations. This proved to be a very effective tool for identifying the optimum technology to produce a drug product.

## 1. Introduction

The continuous manufacturing (CM) of pharmaceutical products is becoming a new technological standard in the pharmaceutical industry. This technology is very promising, so such regulatory bodies as the US Food and Drug Administration (FDA) and the European Medicines Agency (EMA) support its development and implementation as a way to modernize pharmaceutical manufacturing and improve product quality [1,2]. Today, most pharmaceutical products are produced by a method known as batch manufacturing in which different operations, such as weighing, mixing, tableting, or encapsulation, are performed in separate rooms and at different times. This is a long, multi-step process that requires large-scale equipment. CM is faster than batch processing and easier to scale. It eliminates the need to stop and move intermediates between production steps, which can potentially improve the quality of the final pharmaceutical product. Therefore, leaders in the pharmaceutical industry are interested in this technology, which compared to the traditional manufacturing process, speeds up the production cycle, improves overall efficiency, and significantly reduces costs [3,4].

Of the various granulation techniques, twin-screw granulation (TSG) appears to be the most flexible and efficient for CM processes [5,6,7]. It enables the continuous feeding of raw materials, their processing, and uninterrupted transfer to subsequent unit processes. Product quality attributes can be adjusted and optimized just by changing the TSG settings and/or process parameters. The amount of manufactured product can be easily regulated by shortening or lengthening the processing time [8,9,10]. Compared to other granulation equipment, TSG offers the possibility of true continuous production. Very often, processes with the use of a high-shear granulator (HSG) or fluid-bed granulator (FBG) are run in a so-called mini-batches mode. Strictly speaking, these processes are, therefore, rather semi-continuous but are still covered by the relevant ICH or FDA guidelines [1,2].

In this study, a twin-screw granulator was utilized as a continuous processor in the production of liquisolid tablets containing either simethicone or a combination of simethicone and loperamide hydrochloride.

Simethicone is a mixture of polydimethylsiloxane and silicon dioxide and is commonly used as a defoaming agent to relieve abdominal discomfort caused by excessive intestinal gas (e.g., bloating, cramping, flatulence). Simethicone occurs as a hydrophobic, semiopaque, viscous fluid [11]. Loperamide is an effective antidiarrheal drug that is widely administered for the symptomatic treatment of various types of diarrhea [12,13]. The combination of loperamide and simethicone is used to treat acute diarrhea combined with abdominal distension and pain. Loperamide and simethicone show synergistic effects, and their combination is more effective in relieving acute diarrhea than when taken alone [14,15,16]. Such combinations containing 2 mg of loperamide hydrochloride and 125 mg of simethicone are already available on the market [17,18].

The liquisolid technique consists of absorbing a drug in a liquid state by a porous carrier and converting it into a dry, free-flowing, non-adherent powder that is suitable for tableting [19,20]. So, the liquid drug must first be incorporated into the porous structure of a carrier by both absorption and adsorption (collectively referred to as sorption). Thus, the properties of a carrier material, including an elevated specific surface area (SSA) and high loading capacity, play a major role in the preparation of a dry liquid-loaded powder. In pharmaceutical technology, excipients such as magnesium aluminometasilicate, dibasic calcium phosphate, mesoporous calcium silicate, mesoporous magnesium carbonate, or mesoporous silica are often used as carrier material. For some water-insoluble carriers, the incomplete release of certain drugs and even the possibility of drug re-adsorption on the carrier surface have been observed, which may limit their use in solid oral drug formulations [21,22,23]. It has been reported that this technique requires the additional use of very fine and highly adsorptive materials such as silicon dioxide, calcium silicate, or magnesium aluminometasilicates [24,25,26,27,28].

The liquisolid technique is quite versatile, and many examples of its use for the preparation of oral solid dosage forms can be found in the scientific literature. Limpongsa et al. demonstrated its successful application for the preparation of directly compressible orally disintegrating tablets containing hydrophobic cannabinoids [29]. In their study, Kostelanská et al. demonstrated the increased solubility and dissolution rate of liquisolid tablets containing plant extracts [30], and Devi et al. evaluated the feasibility of liquisolid formulations to improve the dissolution rate of ketoprofen and, thus, its bioavailability [31]. Other papers have also reported the applicability of this technology to improve the solubility of poorly soluble drug substances such as triclabendazole or mirtazapine [32,33].

The main objective of the present study was to investigate the possibility of changing the production of liquisolid tablets containing simethicone (as well as a combination of simethicone and loperamide hydrochloride) from a batch process to a continuous one. The continuous process involved preparing liquid-loaded powders using TSG and directly compressing them into tablets on a rotary tablet press. The device used (Thermo Scientific^TM^ Pharma 16 Twin-Screw Extruder, Thermo Fisher Scientific, Karlsruhe, Germany) enabled the simultaneous and continuous feeding of all formulation components (both the liquid and powders). A schematic diagram of the equipment set used in the present study is shown in Figure 1.

The composition and concept of the method for the preparation of simethicone tablets have already been described elsewhere [34,35], although that was a batch-based process with the use of HSG. In the present study, a special porous grade of tribasic calcium phosphate (TRI-CAFOS^®^ 500, Budenheim, Germany) was used as the carrier material. TRI-CAFOS^®^ 500 has a median particle size of around 100 µm, a bulk density of approx. 500 g/L, and a specific surface of approx 85 m^2^/g. An advantage, in the context of the application in question, is its solubility in an acidic environment similar to that prevailing in the stomach [36].

In the course of the research, it examined how the different parameters of the TSG process affected the properties of liquid-loaded powders and the tablets produced from them. The physical properties of the powders and tablets were analyzed and compared using standard compendial methods. In addition, the formulations developed here were compared in performance tests such as disintegration time and the uniformity of dosage units. It should be noted that for loperamide hydrochloride tablets, the pharmacopeial monograph included a dissolution study, whereas, for simethicone tablets, there was only a disintegration test, which did not provide a complete picture of the product’s performance. For this reason, the study investigated defoaming activity as an indicator to compare the efficacy of fabricated tablets.

The distribution of individual components in the fabricated liquisolid tablets was analyzed by chemical imaging using Raman spectroscopy. The inelastic scattering of laser light by molecular vibrations allows the investigation of changes in the vibrational pattern and, thus, the identification of the chemical compound in laser focus [37]. Combined with micrometer mapping capabilities, this enables the visualization of the distribution of formulation components after tableting and facilitates a selection of optimal parameters during processing by comparing their particle size, distribution, and chemical signatures [38].

## 2. Materials and Methods

Simethicone Q7-2243 LVA from Dow Corning GmbH (Wiesbaden, Germany). Loperamide hydrochloride from Vasudha Pharma Chem Limited (Hyderabad, India). Tribasic calcium phosphate, TCP (TRI-CAFOS^®^ 500), and anhydrous dibasic calcium phosphate, DCPA (DI-CAFOS^®^ A150) from Chemische Fabrik Budenheim KG (Budenheim, Germany). Croscarmellose sodium (Ac-Di-Sol^®^ SD-711 NF) from FMC BioPolymer (Philadelphia, PA, USA) and magnesium stearate (Ligamed^®^ MF-2-V) from Peter Greven Fett-Chemi (Venlo, The Netherlands).

### 2.1. Preparation of Liquid-Loaded Powders with Simethicone and Simethicone with Loperamide Hydrochloride

The entire process was carried out in a Thermo Scientific^TM^ Pharma 16 Twin-Screw Extruder (Thermo Fisher Scientific, Karlsruhe, Germany), which was configured so that all ingredients were dosed simultaneously and continuously in the sequence shown in Figure 1. The system operated in twin-screw granulation mode, with an open discharge at the barrel end and screws of 40 ¾ D length. Liquid-loaded powders have the compositions shown in Table 1 and Table 2 and were prepared using different TSG process parameters, as summarized in Table 3. In the case of the formulation with loperamide hydrochloride, it was initially dispersed in simethicone and then loaded onto the carrier. This was conducted due to its low content (0.27% *w*/*w*) to ensure uniform distribution in the liquid-loaded powders, thereby ensuring the required uniformity of dosage units. Figure 2 shows the two screw configurations applied in the study. Powder components were dosed gravimetrically, while simethicone was dosed using a peristaltic pump to achieve the desired process throughput (see Table 3).

The study also compared the effects of different technological processes on the properties of the resulting liquid-loaded powders and subsequently compressed tablets. For this purpose, a powder containing simethicone of the composition given in Table 1 was prepared in a Diosna P1-6 high-shear mixer (Diosna Dierks & Söhne GmbH, Osnabrück, Germany). A detailed description of the process has already been reported elsewhere [34,35]. In addition, the impact of the presence of simethicone on the properties of the powders was checked. For this purpose, a powder mixture containing only excipients (without simethicone) in the ratio shown in Table 1 was prepared by mixing them in a Turbula^®^ T2C mixer (Willy A. Bachofen AG, Muttenz, Switzerland) at 32 rpm for 10 min (placebo blend).

### 2.2. Tableting of the Liquid-Loaded Powders

After leaving the TSG, the liquid-loaded powders were no longer processed and compressed into tablets with a Fette 102i rotary tablet press (Fette Compacting GmbH, Schwarzenbek, Germany) run at a 62.5 rpm press speed. The 12 mm round, flat-faced bevel-edged punches with a break line (Fette Compacting GmbH, Schwarzenbek, Germany) were used. To characterize the tablet ability of the simethicone-containing powders, they were compressed under increasing compaction forces of 10 kN, 15 kN, 20 kN, and 25 kN (equivalent to 87 MPa, 130 MPa, 174 MPa, and 217 MPa, respectively) with 10% precompression. The highest compression force permitted for the punches used was 25 kN.

### 2.3. Analysis of Powders

The flow properties of the powders (liquid-loaded powder and the placebo blend) were characterized using the angle of repose method in accordance with the USP/NF chapter 〈1174〉 “Powder Flow”.

The particle size distribution (PSD) was analyzed with a dynamic image analysis system Camsizer X2 (Retsch Technology GmbH, Haan, Germany) equipped with an X-Jet module, which dispersed the samples using a Venturi nozzle operating at an air pressure of 0.5 bar.

### 2.4. Analysis of Physical Properties of Liquisolid Tablets

Tablets hardness (expressed as breaking force) was analyzed using a Semi-Automatic Tablet Testing System SmartTest 50 (Sotax AG, Aesch, Switzerland). Averages were calculated based on the analysis of 10 randomly selected tablets.

Friability was tested with a friability tester Friabilator (USP) EF-2 (Electrolab, Mumbai, India). The number of tablets corresponding to 6.5 g were weighed and tested at a speed of 25 rpm for 4 min. The tablets were weighed again, and the mass was compared with their initial weight.

The disintegration test was carried out with an SDx-01 disintegration tester (Secom GmbH, Hamburg, Germany) in 900 mL of 0.1 M HCl at 37 °C. Disintegration times of 6 individual tablets were recorded.

### 2.5. Uniformity of Dosage Units and Defoaming Activity of Simethicone Contained in Liquisolid Tablets

The uniformity of dosage units was assessed according to USP/NF Chapter <905> “Uniformity of Dosage Units” based on the analysis of 10 individual tablets with the maximum acceptance value (AV) set at L1 = 15.0. For the determination of simethicone content, the method described in the USP/NF monograph for simethicone tablets was adopted. A number of powdered tablets equivalent to 50 mg of simethicone were transferred to a 100 mL volumetric flask. A total of 25 mL of toluene was added and shaken for 15 min. Then, 50 mL of a 0.1 M HCl solution was added and shaken for another 45 min. On completion of the extraction, 5 mL of the upper layer was taken, and 0.5 g of anhydrous sodium sulfate was added and centrifuged. The resulting samples were analyzed by Fourier transform infrared spectroscopy (FTIR) at a wavenumber of 1260 cm^−1^. A calibration curve was prepared by making a series of dilutions in the stock solution (20 mg/mL of simethicone in toluene) in a concentration range of 0.2–4.0 mg/mL. The determined R2 correlation coefficient stood at 0.9995.

The USP/NF monograph for simethicone tablets provided only one performance test, which was the disintegration time. In the present study, the efficiency of the fabricated tablets was evaluated using a defoaming activity test analogous to that described in the pharmacopeial monograph for simethicone. It involved measuring the time of the collapse of a well-defined foam of a nonionic surfactant, octoxynol-9.

First, a foaming solution containing 1 g of octoxynol-9 in 100 mL of water was placed in a 250 mL glass jar. One tablet was placed in this solution, and the jar was capped. After 5 min, when the tablet had disintegrated, the whole was shaken vigorously for 10 s. The defoaming activity time was determined by recording the time taken for the foam to collapse.

### 2.6. Uniformity of Dosage Units and Dissolution Tests for Loperamide Hydrochloride

The dissolution test was performed in 0.1 N hydrochloric acid using a basket apparatus PTWS 820D (Pharma Test Apparatebau AG, Hainburg, Germany) set at 100 rpm and a temperature of 37 °C. Samples were analyzed using a UV-Vis spectrophotometer T70 (PG Instruments Ltd., Leicestershire, UK) equipped with cuvettes with an optical path length of 10 mm at a detection wavelength of 220 nm. The analysis took 60 min with sample acquisition times 5, 10, 15, 20, 30, and 45 min. A calibration curve was prepared by measuring the absorbance of a series of solutions in the concentration range of 1–5 µg/mL. The determined R2 correlation coefficient stood at 0.9997. An analogous methodology was used to test the uniformity of the dosage units. One tablet was placed in a 1000 mL volumetric flask, 0.1 N hydrochloric acid was added, and the flask was put in an ultrasonic bath for 30 min. After filling the volumetric flask to the line with hydrochloric acid and mixing, the obtained solution was analyzed by the UV/Vis method as described beforehand.

### 2.7. Raman Imaging of Liquisolid Tablets

A DXR™3xi Raman-Imaging-Microscope (Thermo Fisher Scientific, Madison, WI, USA) equipped with an electron-multiplied CCD and a 532 nm laser providing 40 mW power at the sample was used in the present study. Raman mapping was carried out with a 25 µm lateral resolution using an Olympus 10× MPLN objective and with a 5 µm resolution using an Olympus 50× LMPLFLN objective. Raman spectra were collected at 400 Hz in 40 repetitions. The data collection time was around 6 h for a 12 × 12 mm^2^ map registered with a spatial resolution of 25 µm and around 1 h for a 1 × 1 mm^2^ map registered with a spatial resolution of 5 µm. A Multivariate Curve Resolution (MCR) was applied as a chemometric method with background subtraction and 5 estimated compounds to allow the algorithm to find the individual five components included in the liquisolid tablets, i.e., simethicone, TCP, DCPA, croscarmellose sodium, and magnesium stearate. For tablets labeled as “Simethicone/Loperamide TSG”, loperamide hydrochloride was added as the sixth component, and the MCR algorithm changed accordingly.

## 3. Results

### 3.1. Analysis of Powders

Figure 3A shows comparisons of the flow properties expressed by the angle of repose for the placebo blend and liquid-loaded powders prepared with the use of different techniques (i.e., HSG and TSG). Figure 3B summarizes the volume distribution of particle sizes (in µm) that were carried out for the same powders. The individual columns show the average results of the three independent determinations, and the standard deviations are marked with error bars.

### 3.2. Analysis of Physical Properties of Liquisolid Tablets

Figure 4A–C shows the results of the analysis of the physical properties of liquisolid tablets obtained by compressing the liquid-loaded powders using four compression forces, i.e., 10 kN, 15 kN, 20 kN, and 25 kN.

The tablet friability results presented in Figure 4B are shown only up to the Pharmacopoeia acceptance level of 1% weight loss. In the case of tablets made from powder prepared by the HSG process (“Simethicone HSG”) as well as from one of the powders prepared by the TSG process (“Simethicone TSG_3”), tablets compressed with the highest compression force of 25 kN showed capping issues. This resulted in their very high friability results. In the case of tablets labeled as “Simethicone TSG_1”, during the friability test, they lost their upper and lower parts (analogous to the capping problem). In this case, the friability reached values from around 10% to even 30%.

### 3.3. Dissolution Rate of Loperamide Hydrochloride

Figure 5 shows a comparison of the dissolution rate of loperamide hydrochloride from liquisolid tablets obtained by compressing the liquid-loaded powders labeled as “Simethicone/Loperamide TSG” and using four compression forces, i.e., 10 kN, 15 kN, 20 kN, and 25 kN.

### 3.4. Uniformity of Dosage Units and Defoaming Activity of Simethicone Contained in Liquisolid Tablets

The two performance tests, i.e., the uniformity of dosage units for both active ingredients and defoaming activity, were determined only for liquisolid tablets fabricated at 20 kN. This choice was based on the analysis of the physical properties of the tablets shown in Figure 3. For this compression force, for four of the five formulations, the mechanical properties of the fabricated tablets were optimal. At the higher force for some formulations, capping, and high friability were observed. Figure 6A,B shows a comparison of the results for the liquisolid tablets of this study. In the case of formulation labeled as “Simethicone/Loperamide TSG”, two green bars represent the acceptance values (AV) calculated separately for simethicone and loperamide hydrochloride.

### 3.5. Raman Imaging of Liquisolid Tablets

Figure 7 shows the spectral maps recorded on the liquisolid tablets prepared in this study. The visualization of the distribution of individual components was carried out by means of chemometric methods using the MCR algorithm, identifying up to six components. The dark blue color represents a mixture of simethicone and TCP and light blue—pure TCP, red—DCPA, green—magnesium stearate, orange—croscarmellose sodium, and pink (for “Simethicone/Loperamide TSG” formulation only)—loperamide hydrochloride.

To the left are Raman maps of the entire tablets recorded at a spatial resolution of 25 µm. In two cases, the maps of only half of the tablets were registered but they can be considered very representative. On the right are detailed spectral images of 1 × 1 mm^2^ tablet fragments recorded with a spatial resolution of 5 µm.

The areas of the spectral maps marked in dark blue indicate regions where the superposition of the Raman signatures of TCP and simethicone were recorded (i.e., spectral signatures containing the combined bands characteristic for both components, as shown in Figure 8). The light blue color indicates sites where the spectra of pure TCP were recorded. At the same time, spectral signatures, characteristic of pure simethicone, were not detected. The areas where the Raman spectra characteristic of DCPA was recorded are marked in red. In some locations, these spectra were not pure but included spectral signatures of simethicone as well. This indicates a local release of simethicone from TCP particles during compression, which was reabsorbed by DCPA particles.

Figure 9 shows the Raman correlation map of the “Simethicone/Loperamide TSG” formulation. Red spots indicate areas of high correlation for Raman spectra with that of loperamide hydrochloride. Loperamide hydrochloride is distributed evenly in the tablet but variations in particle size are evident. This suggests the inhomogeneous particle size distribution of the drug substance. Larger crystals of loperamide hydrochloride were unable to break up in simethicone under the conditions of the process before being deposited on the carrier particles.

## 4. Discussion

The main goal of this study was to investigate the possibility of converting the production of liquisolid tablets containing 125 mg of simethicone, as well as a combination of 125 mg of simethicone and 2 mg of loperamide hydrochloride, from a batch process to a continuous one. The first one, based on high-shear granulation, was described in earlier work [34]. The continuous process employed here involved the preparation of liquid-loaded powders using TSG, followed by compressing them into tablets on a rotary tablet press (see Figure 1).

Simethicone is a liquid, oily substance, and to form a dry, free-flowing powder that can be further compressed into tablets, it must first be absorbed in a suitable carrier. These are usually porous substances with an elevated specific surface area and high sorption capacity [24,39]. There have been numerous works reporting the difficulties in achieving the sufficient flowability of simethicone-containing powders, as well as the satisfactory compatibility of the final blends for compression. Moreover, simethicone is a thick liquid characterized by high viscosity, so the challenge is to ensure that it is evenly distributed in the dry powder, as well as to load it onto the porous carrier in the required amount so that it is not squeezed out during tableting. In addition, during compression, there is a local increase in temperature, which can cause the simethicone to melt, resulting in the tablet mass sticking to the punches [40,41,42].

In this study, porous TCP (TRI-CAFOS^®^ 500) was used as a carrier on which simethicone was deposited. This grade of TCP was chosen because of its high specific surface area (about 85 m^2^/g) as well as its solubility in acidic conditions [36]. These properties, on the one hand, facilitated the efficient sorption of simethicone, while on the other hand, promoted the release of the active ingredient in the acidic environment of the stomach.

The carrier with simethicone absorbed in it did not show sufficient compatibility. Therefore, to increase the mechanical strength of the tablets, a directly compressible grade of DCPA (DI-CAFOS^®^ A150) was employed as a compaction aid. The flowability of the liquid-loaded powders (final tableting blends), compared to the placebo blend (without simethicone), was degraded from good/excellent to varying degrees, depending on the preparation process (see Figure 3A). The best flow properties (fair/good) were obtained for the formulation prepared in HSG (labeled as “Simethicone HSG”) and for the formulation prepared in TSG with low throughput and the screw configuration “A” with one kneading zone (“Simethicone TSG_1”). This can be attributed to the larger particle sizes measured for these formulations, as shown in Figure 3B. The particle dimensions were noticeably larger than those of the placebo blend, indicating that agglomerates were formed during the processing of powders. The liquid-loaded powders of formulations labeled as “Simethicone TSG_2“, “Simethicone TSG_3”, and “Simethicone/Loperamide TSG“ were prepared using the “B” screw configuration. The use of two kneading zones prevented the formation of agglomerates and even the partial fragmentation of the original particles, as reflected by the formation of a very fine particle fraction. This negatively affected the powder flow of the final tableting blends, which according to the pharmacopoeia, could be described as passable (formulations “Simethicone TSG_3” and “Simethicone/Loperamide TSG”) and even poor for formulation “Simethicone TSG_2”. In this context, faster powder processing seems to be more beneficial. Nevertheless, the flow properties of all liquid-loaded powders, regardless of their preparation method, were sufficient to be effectively compressed into tablets using a rotary tablet press fitted with a force feeder, which is a standard component of contemporary tablet pressing machines.

The mechanical strength of the tablets depended on the method of manufacturing the liquid-loaded powders (see Figure 4A,B). By far, the least favorable properties were exhibited by the formulation labeled as “Simethicone TSG_1”, for which the tablet hardness (breaking force) decreased with increasing compaction force. The liquisolid tablets produced showed, in this case, a very high capping/lamination tendency, and in the friability test, they broke up into smaller fragments. The change in the screw configuration has improved the tableting properties of the liquid-loaded powders and enabled the production of tablets with noticeably better hardness and very low friability. An interesting phenomenon was observed for the formulations labeled as “Simethicone HSG“ and “Simethicone TSG_3”, for which the application of the highest compression force of 25 kN resulted in a capping problem and a noticeable reduction in tablet hardness and friability. Liquid-loaded powder labeled as “Simethicone TSG_3” was prepared with higher throughput than “Simethicone TSG_2”, for which this problem did not appear. It can be assumed that when the powders passed through the TSG faster, the pores of the carrier were not completely filled by the simethicone, and the air remained entrapped. A similar phenomenon probably occurred during the preparation of the powder in the HSG. When compressed with a higher force, the air was released from the pores, which resulted in capping. Interestingly, in the case of a formulation containing loperamide hydrochloride in addition to simethicone (labeled as “Simethicone/Loperamide TSG”), which was prepared in identical conditions to the formulation “Simethicone TSG_3”, no tablet capping occurred, and it exhibited the best tableting properties. This indicates that even such a small amount of the drug substance (0.27% *w*/*w*) can noticeably improve the tableting properties of powder mixtures.

In a series of performance tests, the prepared tablets showed very good effectiveness. The tablet disintegration times for all formulations prepared in this study were many times faster than those required by the USP/NF monograph for Simethicone Tablets, i.e., 30 min, which amounted to less than 4 min (see Figure 4C). Disintegration times were clearly correlated with the hardness of the liquisolid tablets and were close to each other, although fastest for the formulation labeled “Simethicone/Loperamide TSG”. Similarly, the defoaming activity time of the tablets was very fast. After the tablet disintegrated, simethicone showed its effect for less than 1.5 min in all cases (see Figure 6B). This shows that, despite the differences in the physical properties of the liquid-loaded powders and compressed liquisolid tablets, the performance of all formulations remained at a similar level.

For tablets marked “Simethicone/Loperamide TSG”, dissolution studies were carried out for loperamide hydrochloride. The tablets showed a very rapid release of the active ingredient, with more than 85% of the substance dissolved within the first 15 min of the test (see Figure 5). According to the guidelines, the similarity of dissolution profiles can be assumed for such rapid release without further mathematical evaluation [43,44]. Nonetheless, it can be observed that the rate of release for loperamide hydrochloride slowed down slightly as the compression force, and thus the hardness of the tablets, increased.

The uniformity of dosage units test showed satisfactory results for both active ingredients contained in liquisolid tablets prepared in this study (see Figure 6A). Although differences in the calculated AV values for simethicone could be seen for different formulations, it should be noted that they were all well below the pharmacopoeia requirement of L1 = 15.0 (see USP/NF Chapter <905> “Uniformity of Dosage Units”). Since loperamide hydrochloride is present in liquisolid tablets of the formulation labeled “Simethicone/Loperamide TSG” in a very small amount (0.27% *w*/*w*), it can be considered the most critical ingredient in terms of maintaining an adequate level of content uniformity. Nevertheless, even in this case, the AV value did not exceed 7seven. The results obtained confirm that the continuous method for the preparation of liquisolid tablets proposed in this work using TSG allowed the production of dosage forms which have satisfactory physical properties and the required quality attributes.

Raman maps (see Figure 7) evaluated by chemometric methods allowed for the chemical profiling of liquisolid tablets. The analysis did not reveal the presence of free simethicone, which was completely adsorbed, mainly on porous TCP (dark blue areas that make up the bulk of the maps). During tableting at higher compression forces, the local breaking and crushing of TCP particles occurred, leading to the release of small amounts of simethicone. The resorption of simethicone by DCPA particles further prevented its’ squeezing out during tableting, as can be evidenced by the superposition of the Raman signatures of simethicone and DCPA in the MCR component spectrum (analogous to the case of TCP and simethicone shown in Figure 8).

An analysis of the red areas in the Raman maps in Figure 7, assigned to DCPA (and traces of simethicone), indicates how the different ways of preparing liquid-loaded powders affect the distribution of components in liquisolid tablets. It strongly depends on the type of granulator used (HSG or TSG) and, in the case of TSG, on the configuration of the screws and the speed of the process (i.e., the speed at which the powders were passed through the TSG). Formulations labeled as “Simethicone TSG_1” and “Simethicone TSG_2” were produced at the same, low throughput of 5 kg/h. In the case of the first formulation, the screw configuration included only one kneading zone, so the powder mainly resided in the mixing area. This led to the formation of larger agglomerates, which are represented by large red areas on the Raman maps. These observations correlate very well with the results of the particle size analysis, where the largest particles were measured for this formulation (see Figure 3B). In the case of the second formulation, the screw configuration included two kneading zones. With this setting, the particle size distribution was narrowed, and the distribution of ingredients in the tablet was improved (compared with Figure 3B and Figure 7).

In the Raman maps of the “Simethicone TSG_3” formulation, which was produced at a higher throughput of 10 kg/h, the DCPA particles appeared to be slightly larger in size (even though PSD analysis did not reveal increased particle dimensions). Nevertheless, the particles were distributed very evenly throughout the tablet. A formulation containing loperamide hydrochloride (“Simethicone/Loperamide TSG”) was also produced with the “B” screw configuration at a higher process throughput of 10 kg/h. The spectral map shows a uniform distribution of formulation components, but numerous particles of pure TCP were also revealed. This suggests that the higher processing speed may not provide the efficient sorption of simethicone on the carrier particles. It should also be noted that, in the case of this formulation, simethicone additionally contained loperamide hydrochloride, which certainly affected the liquid’s viscosity and the speed of its penetration into the particles of the carrier. Nevertheless, loperamide hydrochloride in liquisolid tablets was evenly distributed, as confirmed by the results of the uniformity of dosage units, which met the acceptance criteria required by the pharmacopoeia.

Raman maps of the formulation prepared by HSG (“Simethicone HSG”) show very large agglomerates of DCPA and magnesium stearate particles, which are relatively uniformly distributed in the tablet (similar conclusions regarding particle sizes can be drawn from PSD analysis). No increased number of pure TCP particles could be seen, suggesting the sufficient sorption of simethicone on the carrier.

## 5. Conclusions

The results of the conducted study show that the employment of TSG for the preparation of liquisolid tablets allows the batch production process using HSG technology to be successfully transformed into a continuous one. The active ingredients used in this study posed a major technological challenge, as simethicone was a liquid, oily substance, and loperamide hydrochloride was used in the formulation in very small quantities. Despite these difficulties, the appropriate choice of carrier and the adjustment of the TSG process parameters enabled the optimization of the properties of the liquid-loaded powders and made it possible to efficiently produce liquisolid tablets with satisfactory physical properties and performance. Raman imaging has proved to be a very powerful analytical technique for evaluating the effect of the production process on the efficiency of simethicone sorption on the carrier, as well as the distribution of the individual components in the tablet.

## Figures and Tables

**Figure 1 pharmaceutics-15-01265-f001:**
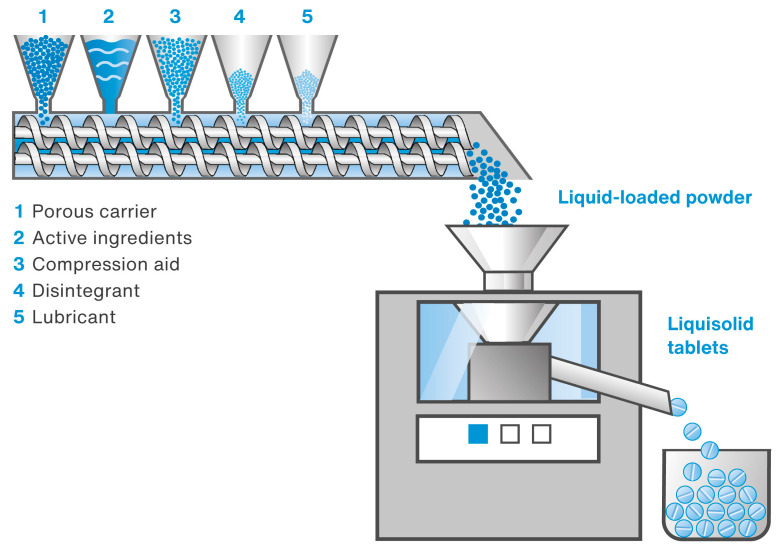
Schematic diagram of a production line for manufacturing liquisolid tablets containing simethicone and loperamide hydrochloride.

**Figure 2 pharmaceutics-15-01265-f002:**
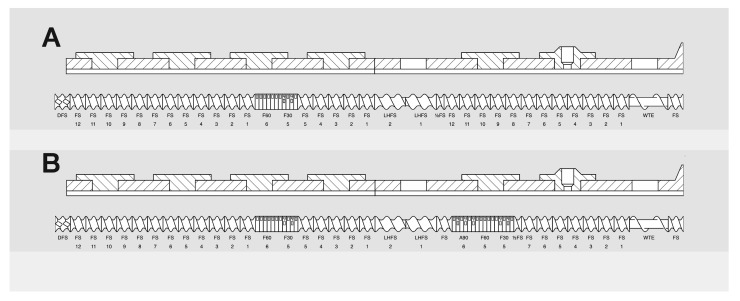
Schematic diagram of the screw configuration with one (**A**) or two (**B**) kneading zones and corresponding barrel set-up (prepared in Twin Screw Configurator software version 1.34 (6) from Thermo Fisher Scientific, Karlsruhe, Germany).

**Figure 3 pharmaceutics-15-01265-f003:**
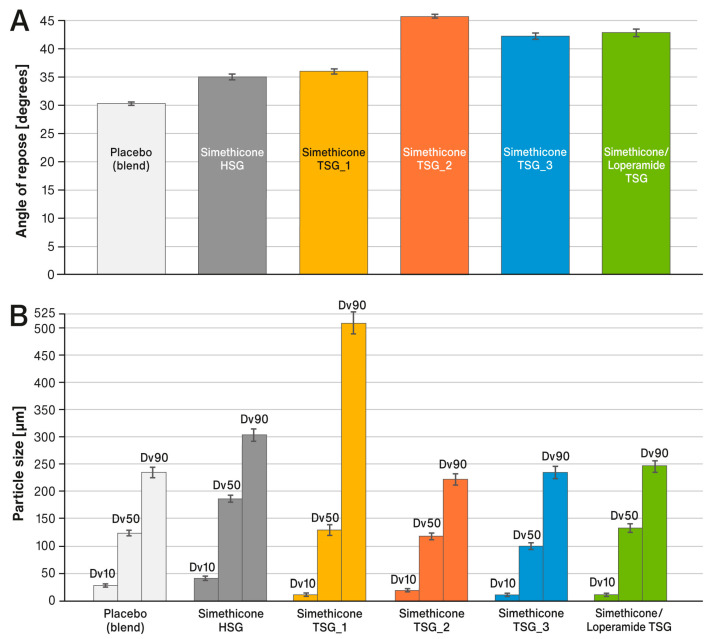
Comparison of (**A**) flow properties (angle of repose) and (**B**) particle size distribution of liquid-loaded powders and placebo blend. Means of *n* = 3; SD is indicated by the error bars.

**Figure 4 pharmaceutics-15-01265-f004:**
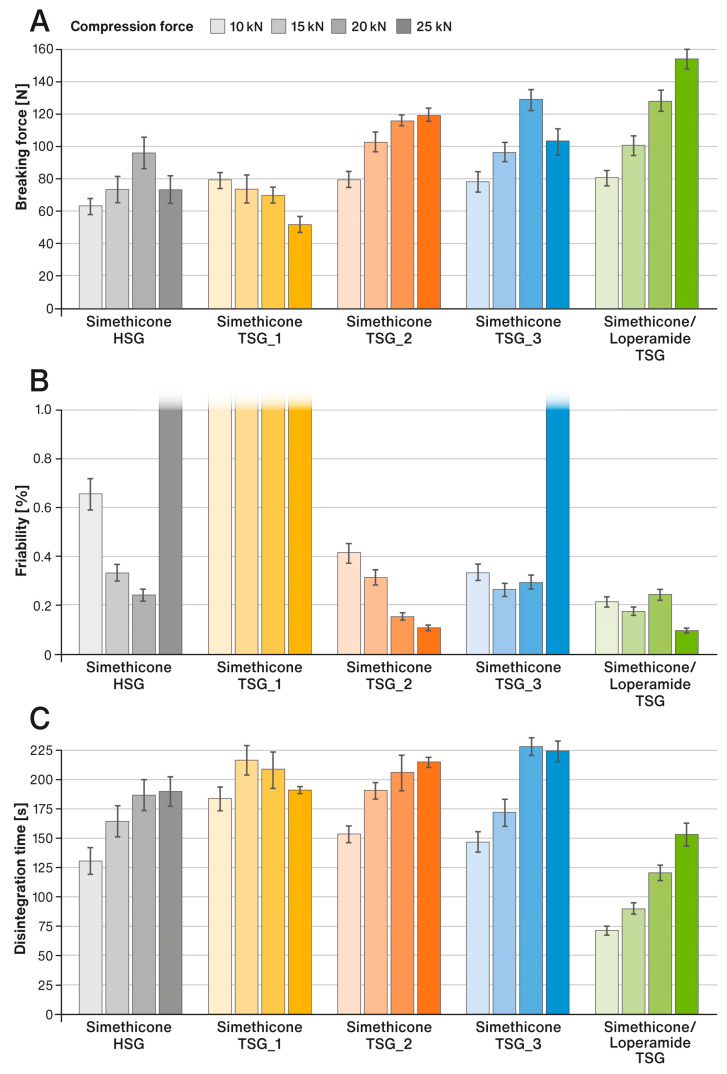
Comparison of (**A**) hardness (breaking force); means of *n* = 10, (**B**) friability; means of *n* = 3, (**C**) disintegration time; means of *n* = 6 of liquisolid formulations compressed under various compression forces. SD is indicated by the error bars.

**Figure 5 pharmaceutics-15-01265-f005:**
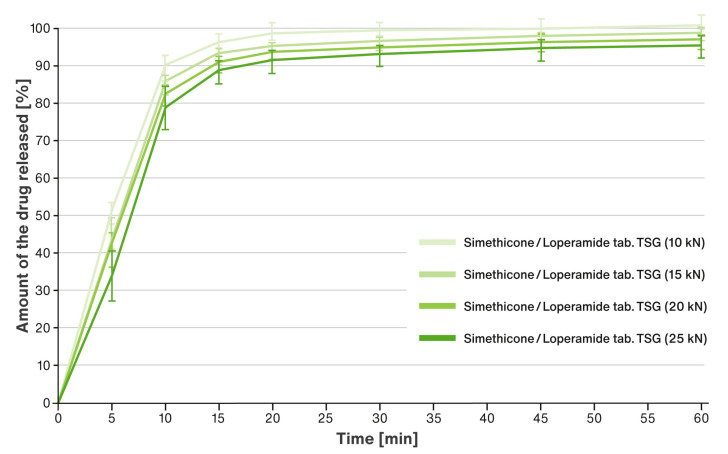
Comparison of dissolution rate of loperamide hydrochloride from liquisolid formulations compressed under various compression forces. Means of *n* = 6; SD is indicated by the error bars.

**Figure 6 pharmaceutics-15-01265-f006:**
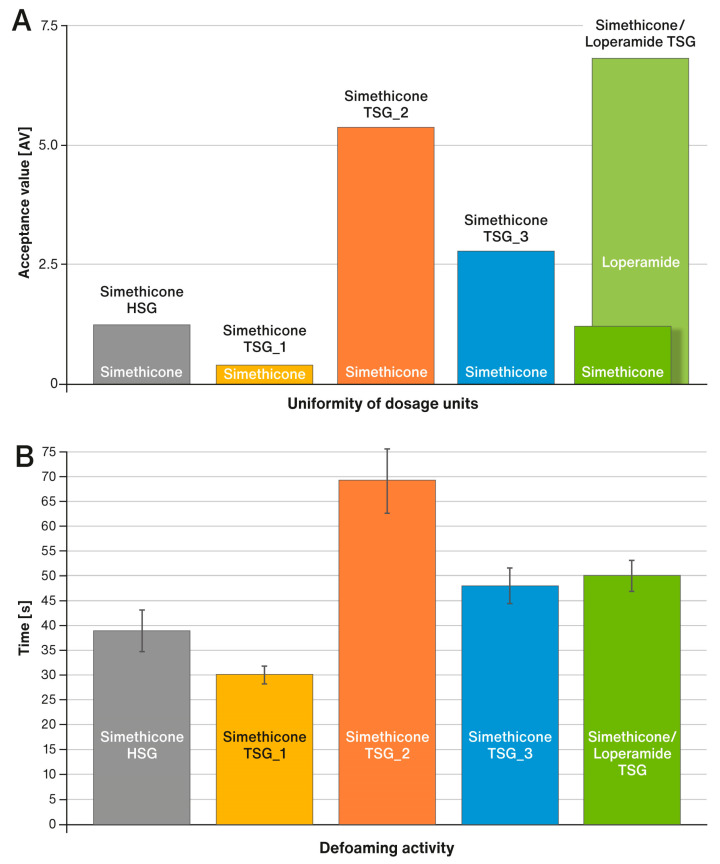
(**A**) Uniformity of dosage units and (**B**) Defoaming activity of liquisolid tablets compressed at 20 kN; means of *n* = 6. SD is indicated by the error bars.

**Figure 7 pharmaceutics-15-01265-f007:**
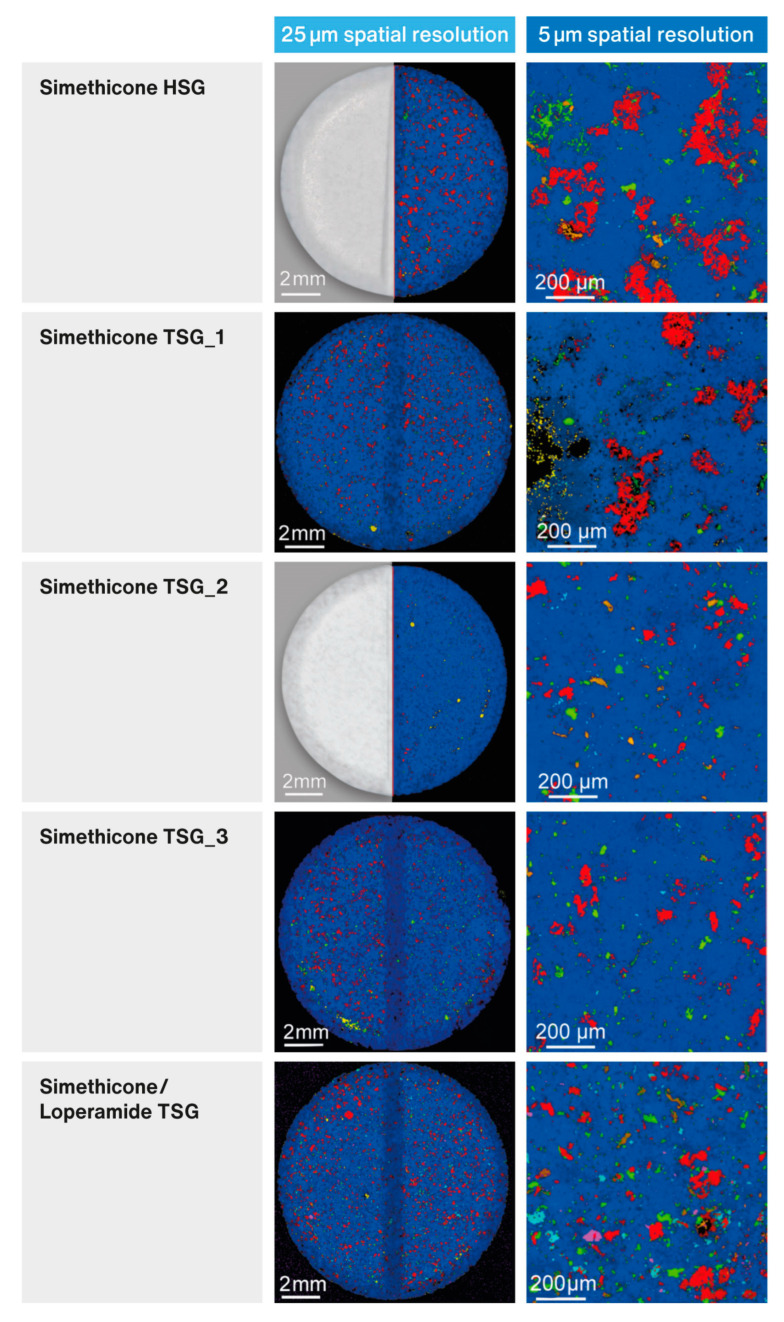
Raman maps of liquisolid tablets registered with 25 µm and 5 µm spatial resolutions. The spectra are separated into chemical components by MCR analysis and displayed in different colors.

**Figure 8 pharmaceutics-15-01265-f008:**
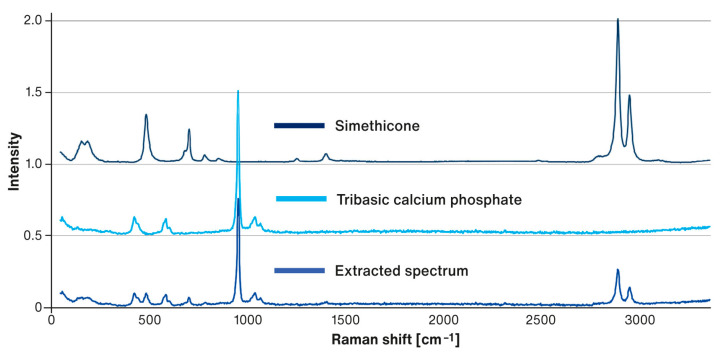
Raman spectra of simethicone, TCP, and an example of a spectrum recorded in the dark blue areas of the Raman correlation maps showing the superposition of individual Raman spectra.

**Figure 9 pharmaceutics-15-01265-f009:**
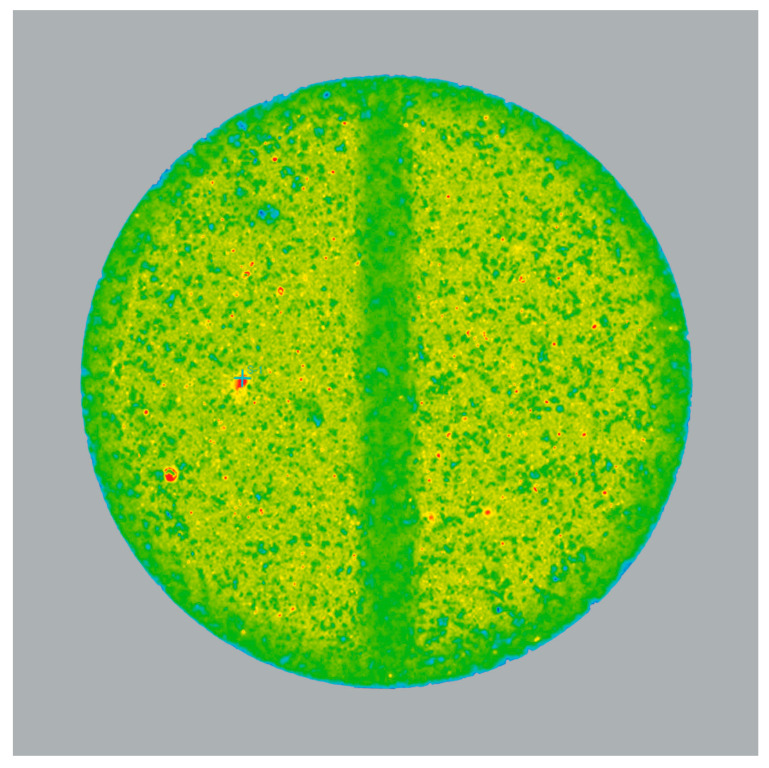
Raman correlation map showing the spatial distribution of loperamide hydrochloride; red sites are those with the highest, and blue sites are those with the lowest correlation of the Raman spectrum with that of loperamide hydrochloride.

**Table 1 pharmaceutics-15-01265-t001:** Composition of simethicone liquisolid tablets.

Ingredient	mg/Tablet	Concentration (% *w*/*w*)	Function
Simethicone Q7-2243 LVA	125.0	17.0	Active ingredient
TRI-CAFOS^®^ 500	374.8	51.0	Carrier
DI-CAFOS^®^ A150	220.4	30.0	Diluent/compression aid
Ac-Di-Sol^®^ SD-711 NF	7.4	1.0	Disintegrant
Ligamed^®^ MF-2-V	7.4	1.0	Lubricant
Total	735.0	100.0	

**Table 2 pharmaceutics-15-01265-t002:** Composition of simethicone and loperamide hydrochloride liquisolid formulations.

Ingredient	mg/Tablet	Concentration (% *w*/*w*)	Function
Loperamide hydrochloride	2.0	0.27	Active ingredient
Simethicone Q7-2243 LVA	125.0	16.96	Active ingredient
TRI-CAFOS^®^ 500	375.1	50.90	Carrier
DI-CAFOS^®^ A150	220.1	29.87	Diluent/compression aid
Ac-Di-Sol^®^ SD-711 NF	7.4	1.00	Disintegrant
Ligamed^®^ MF-2-V	7.4	1.00	Lubricant
Total	737.0	100.0	

**Table 3 pharmaceutics-15-01265-t003:** Various TSG process parameters.

	Simethicone TSG_1	Simethicone TSG_2	Simethicone TSG_3	Simethicone/Loperamide TSG
Screw configuration	A	B	B	B
Screw speed	200 rpm	200 rpm	400 rpm	400 rpm
Total throughput	5 kg/h	5 kg/h	10 kg/h	10 kg/h

## Data Availability

All relevant data/results collected in this study are presented in the article.

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
