# Peer review of "Towards the Continuous Manufacturing of Liquisolid Tablets Containing Simethicone and Loperamide Hydrochloride with the Use of a Twin-Screw Granulator"

_pharmaceutics, 2023, doi:10.3390/pharmaceutics15041265_

Round 1

Reviewer 1 Report

The article deals with continuous manufacturing of liquisolid tablets, containing simethicone and loperamide hydrochloride with the use of a twin-screw granulator.

The Introduction is well structured, containing information about continuous manufacturing, liquisolid technique, the used active pharmaceutical ingredients and describes the main objective of the study. 

The Methods are described in sufficient amount of details and are well chosen.

The Results are presented in logical order and are easy to follow.

A few suggestion for the authors:

In the Introduction part there be useful to add references regarding liquisolid tablets with the used APIs. 

In chapter 2.2 Preparation of liquid-loaded powders with simethicone and loperamide hydrochloride some information about screw configuration B1 and B2 is missing (in lines 447-455 in Discussion appears partially).

Author Response

Together with my co-authors, I would like to thank the Reviewer for valuable comments and suggestions. Below we have included our responses.

In the Introduction part there be useful to add references regarding liquisolid tablets with the used APIs. 

As suggested, relevant information has been added between lines 82-90 of the revised manuscript.

In chapter 2.2 Preparation of liquid-loaded powders with simethicone and loperamide hydrochloride some information about screw configuration B1 and B2 is missing (in lines 447-455 in Discussion appears partially).

In the lines indicated, only formulation optimization is described when using a screw configuration with two kneading zones (B1 vs. B2). This applies only to formulations containing one active ingredient, i.e. simethicone. The formulation with two active substances was produced only with an optimized B2 screw configuration. Information about this is given in Table 3. Similarly, lines 371-373 give information about the use of this configuration for the preparation of the Simethicone/Loperamide TSG formulation.  For clarification, the description in lines 463-465 (of the revised manuscript) was supplemented with the relevant information.

Please note that, as one of the reviewers suggested, the revised manuscript adopted the names of the screw configurations as shown in Figure 2, i.e. A and B

Reviewer 2 Report

1. Make the statement"the liquid drug is embedded in the porous structure of a vector by both absorplion and adsorption" and the in the text you talk only about absorption.

2. Sometimes the text is too descriptive and repetitive, it should be more concise.

3. The sequence of the addition of formulation components in the continuous granulation process is not clearly reported (e.g., addition time)

4. In the figure 2, you describe the configuration of screw A and B, not B1 and B2.

5. To better analyse the interection between formulation variables (active substances and porous carrier) and  process parametrs (screw configuration and screw speed) you had to apply the DoE.

Author Response

Together with my co-authors, I would like to thank the Reviewer for valuable comments and suggestions. Below we have included our responses.

  1. Make the statement "the liquid drug is embedded in the porous structure of a vector by both absorplion and adsorption" and the in the text you talk only about absorption.

The phenomenon in which the processes of adsorption and absorption occur together is called sorption. This is highlighted in line 71 of the revised manuscript. Consistently, this nomenclature was applied later in the manuscript – see lines 346, 441, 468, 478, 490.

Between lines 76-81 the terms “re-adsorption” and “adsorptive materials” have not been changed because they are citations from the related articles.

  1. Sometimes the text is too descriptive and repetitive, it should be more concise.

We have reviewed the manuscript and made the relevant corrections, which are highlighted in red. Some fragments were removed, and some may seem repetitive. However, in the discussion of the results, certain passages on the same aspect of the work are discussed from different perspectives and are difficult to skip. We hope that the revised manuscript will be found acceptable.

  1. The sequence of the addition of formulation components in the continuous granulation process is not clearly reported (e.g., addition time)

The order in which the ingredients are added is shown in Fig. 1. a suitable clarification has been added to the manuscript in lines 139 - 151 of the revised manuscript. Please note that the process was not controlled by the time of addition of the components but by the rotational speed of the screws, which resulted in the intended throughput of the process as given in Table 3.

  1. In the figure 2, you describe the configuration of screw A and B, not B1 and B2.

In the entire manuscript, the relevant corrections were made, and the naming of the "A" and "B" screw configurations was adopted.

  1. To better analyse the interection between formulation variables (active substances and porous carrier) and process parametrs (screw configuration and screw speed) you had to apply the DoE.

Thank you for this comment. Please note that the article does not show the detailed relationships between the various process variables, but only, as the title suggests, the direction of change that makes it possible to turn a batch process into a continuous one. And indeed, as the reviewer suggests, a thorough understanding of the process would require DoE methodology, but this is rather material for another in-depth article, which for the authors is an interesting indication for follow-up research.

Reviewer 3 Report

Daniel and Co-Workers have studied "Towards continuous manufacturing of liquisolid tablets containing simethicone and loperamide hydrochloride with theus e of a twin-screw granulator". The article was well written with good scientific support. However, please address the following comments:

1) Please indicate the rationale for selecting the screw speed and its significance on final drug product

2) Please submit any stability studies or compatability studies conducted for the API's with excipients.

3) Please submit the standard curves along with the data generated for standard curves for both APIs

4) Please provide the impact of having A configuration and B configuration on the final product.

5) Please provide details on how the inhomogeneous particle size distribution of drug substance can be addressed?

Author Response

Together with my co-authors, I would like to thank the Reviewer for valuable comments and suggestions. Below we have included our responses (please see attached pdf file).

1) Please indicate the rationale for selecting the screw speed and its significance on final drug product

The selection of the screw speed is an experimentally determined value and is based on the properties of the materials being processed. For a 10 kg/h feed rate, 200 rpm screw speed would be far too low. This would cause the barrel to fill up too much with material in the kneading zones and we could get into a torque alarm. In addition, in the feed zone, the material would not be taken into the barrel and would build up in the feeding channel. Therefore, at 10 kg/h, the screw speed was increased to 400 rpm.  Furthermore, at 10 kg/h and 200 rpm the specific mechanical energy would be much too low to mix the different materials homogeneously. The choice of 600 rpm showed too much energy input and inhomogeneity of the mixture (particle size distribution) therefore we stayed at 400 rpm in the continuous process.

Regarding the second part of the question, these matters are addressed in the "Discussion" section, among others, in lines 394 – 397 or 460 – 471 of the revised manuscript.

2) Please submit any stability studies or compatability studies conducted for the API's with excipients.

Please take into consideration that, as the title suggests, the purpose of the presented research was to demonstrate the feasibility of converting a batch production process to a continuous one. Unfortunately, the plan for this research did not include drug-excipient compatibility studies or stability studies.

3) Please submit the standard curves along with the data generated for standard curves for both APIs

Calibration curve data for UV/Vis analysis according to section 2.6.:

  (please see attached pdf file)

Calibration curve data for FTIR analysis according to section 2.5.:

(please see attached pdf file).

4) Please provide the impact of having A configuration and B configuration on the final product.

Please note that these issues have been addressed in the "Discussion" section, among others, in lines 365 – 378, 3865 – 390, or 447 – 459 of the revised manuscript.

5) Please provide details on how the inhomogeneous particle size distribution of drug substance can be addressed?

Please note that despite some variation in the particle size of loperamide hydrochloride, it was evenly distributed inside the tablets (see Figure 9) and also between all liquisolid tablets of this study, which was further confirmed by the uniformity of dosage units test. In the context of the results obtained, there was no need for special processing of this drug substance. The procedure employed was sufficient to achieve the required quality attributes of the liquisolid tablets.

Reviewer 4 Report

Manuscript ID: pharmaceutics—2308849

Title: Towards continuous manufacturing of liquisolid tablets containing simethicone and loperamide hydrochloride with the use of a twin-screw granulator

Reviewer comments

Generally, the articles discuss a good point. The target of drug production is to provide the patients with safe and effective product. Manufacturing using process that produce products with less batch to batch variation will be more economic and minimize the risk of failure in different steps. Moreover, the author study the effect of critical process parameters that can affect the product quality. The writing style was smooth. The figures were self-explaining.

·        Table 3 is mentioned in the text before table 2, please rearrange

·        Page 4, line 152, “In this case, the loperamide”. Please replace with In this case, loperamide

·        In section 2.7, the samples was analyzed spectrophotometrically at λ 220, in which a lot of interference could occurred at this low wavelength. How could be the author sure that there is no interference.

·        In figure 2, please identify A, B in the caption.

Author Response

Together with my co-authors, I would like to thank the Reviewer for valuable comments and suggestions. Below we have included our responses (please see attached pdf file).

Table 3 is mentioned in the text before table 2, please rearrange

Please see the changes made between lines 137 - 168 in the revised manuscript.

Page 4, line 152, “In this case, the loperamide”. Please replace with In this case, loperamide

The relevant changes have been made to the manuscript.

In section 2.7, the samples was analyzed spectrophotometrically at λ 220, in which a lot of interference could occurred at this low wavelength. How could be the author sure that there is no interference.

Before starting the analysis, the method was checked, among other things, for its specificity. The figure shown below provides a comparison of 3 independent spectra of solutions of liquisolid tablets containing loperamide hydrochloride and solutions of liquisolid tablets without this drug substance. As can be seen in this case there is no interference between the maxima of the spectra at a wavelength of 220 nm. Please note that often the determination of loperamide is carried out even at a lower wavelength, i.e. 214 nm. In our research we suggested the monograph of loperamide hydrochloride from Ph.Eur. 9.0 as well as the works of some authors, including Shu et al. (https://doi.org/10.1016/j.ajps.2017.08.009), Ruddy et al. (Acta Pol Pharm. 2002, 59(1), 15-18), Begum et al. (International Journal Of Universal Pharmacy And Bio Sciences 2015, 4(6), 335-345, and others. Although the wavelength used does not represent a clear absorption maximum, the linearity results (please see attached pdf file) were found satisfactory.

In figure 2, please identify A, B in the caption.

The corresponding changes have been made to the caption of Figure 2.
